# Cross-ecosystem carbon flows connecting ecosystems worldwide

Isabelle Gounand [1,2], Chelsea J. Little [1,2], Eric Harvey [1,2,3] & Florian Altermatt [1,2]

Ecosystems are widely interconnected by spatial flows of material, but the overall importance of these flows relative to local ecosystem functioning remains unclear. Here we provide a quantitative synthesis on spatial flows of carbon connecting ecosystems worldwide. Cross-ecosystem flows range over eight orders of magnitude, bringing between $10^{-3}$ and $10^5$ gC m$^{-2}$ year$^{-1}$ to recipient ecosystems. Magnitudes are similar to local fluxes in freshwater and benthic ecosystems, but two to three orders of magnitude lower in terrestrial systems, demonstrating different dependencies on spatial flows among ecosystem types. The strong spatial couplings also indicate that ecosystems are vulnerable to alterations of cross-ecosystem flows. Thus, a reconsideration of ecosystem functioning, including a spatial perspective, is urgently needed.

[1] Department of Evolutionary Biology and Environmental Studies, University of Zurich, Winterthurerstrasse 190, 8057 Zürich, Switzerland. [2] Department of Aquatic Ecology, Eawag: Swiss Federal Institute of Aquatic Science and Technology, Überlandstrasse 133, 8600 Dübendorf, Switzerland. [3] Department of Ecology and Evolutionary Biology, University of Toronto, Toronto M5S 3B2, Canada. Correspondence and requests for materials should be addressed to I.G. (email: isabelle.gounand@eawag.ch) or to F.A. (email: florian.altermatt@eawag.ch)

Ecosystems and the services that they provide are essential for material and cultural human welfare[1–3], but paradoxically, human activities threaten ecosystem integrity[4,5]. Maintaining functional ecosystems, or restoring degraded ones, requires the identification of dominant mechanisms driving their dynamics. At the local scale, ecologists have accumulated extensive data on individual ecosystems' functioning[6]. But ecosystems are not isolated. They are connected by spatial flows of organisms and material. The role of dispersing organisms on large-scale species coexistence and community dynamics is well studied[7,8]. However, it remains unclear to which extent local ecosystem functioning also depends on cross-ecosystem flows of material, such as detritus or nutrients[9–11]. According to the recently developed meta-ecosystem theory[8,12], such cross-ecosystem flows can induce strong interdependencies between ecosystems and drive ecosystem functioning[13–16]. Emblematic cases of resource subsidies moving between ecosystems include passive transports, such as of leaves windblown from forests to streams[17,18], or active transports, such as aquatic insects emerging onto land[19,20]. Some effects of these spatial flows, for example on specific trophic levels[21–23], have been synthesized by comparing effect sizes across different study systems. However, when it comes to quantifying spatial flows at the ecosystem level, the dispersion of data over many research areas and inconsistencies in the measurement units have hitherto precluded a more general and synthetic overview of spatial flows.

Here, we conduct a quantitative synthetic assessment of cross-ecosystem flows of carbon connecting the major ecosystem types across the globe (Fig. 1). Specifically, we compare the magnitudes of cross-ecosystem flows to within-ecosystem fluxes in order to infer their potential contribution to ecosystem functioning. This also provides the basis to identify ecosystems' vulnerability to increasing alterations of spatial flows in the context of ongoing global changes[24]. We based our analysis on generally convertible estimates in units of carbon (gC m$^{-2}$ year$^{-1}$; see Methods section) and systematically searched for quantifications of spatial flows connecting terrestrial (forest, grassland, agro-ecosystem, desert), freshwater (stream, lake), and marine (pelagic and benthic) ecosystems. To put these spatial flows in a relevant context, we also assembled comprehensive quantifications of local biological fluxes of carbon (i.e., gross primary production, ecosystem respiration, and decomposition) within the different ecosystem types, again all converted into gC m$^{-2}$ year$^{-1}$. This allowed us to directly compare the magnitude of local ecosystem fluxes to cross-ecosystem flows. We additionally compiled

measurements of net ecosystem production to relate the importance of spatial flows to the degree of ecosystem heterotrophy. Overall, we assembled 518 measurements of spatial flows and 2516 of local fluxes, totaling 3034 data points extracted from 557 studies. Analyzing this data set with its internally-consistent measurements in carbon units reveals the wide range of magnitude of spatial material flows and their widespread importance to local ecosystem functioning. Specifically, we characterize common ecosystem couplings in which the production of freshwater and some benthic ecosystems depends on carbon exported from terrestrial and pelagic ecosystems, respectively, and thus might be sensitive to alterations of these flows. By contrast, terrestrial and pelagic ecosystems seem more independent from spatial carbon flows. However, we also identify some gaps in flow quantification such as with spatial flows driven by animal movements, which could refine our view of ecosystem couplings.

## Results

**Origins and magnitudes of cross-ecosystem flows**. Cross-ecosystem flows of carbon range over eight orders of magnitude in the mass of carbon annually transported to ecosystems (gC m$^{-2}$ year$^{-1}$), from a few milligrams (aquatic insects deposited in forests) to more than ten tons per meter squared and year (wrack on shores) (Fig. 2). The materials transported among ecosystems are as various as living animals, dead plants and animals, and dissolved carbon. Ecosystems occupying the lowest elevations in landscapes and seascapes receive downward flows of dead material or small organisms from ecosystems above them, while terrestrial ecosystems receive more lateral flows, for instance of material transported by wind, ocean tides, or animal movements. Not surprisingly, ecosystems dominated by primary producer biomass (e.g., forests, grasslands, submarine meadows, kelp forests), export the largest flows of primary production-derived material (median/interquartile range [IQR] of 148.7/[47.1–246.6] and 164/[93.2–1659.3] gC m$^{-2}$ year$^{-1}$ for terrestrial plants and macro-algae, respectively). By comparison, invertebrate subsidies are substantially lower (median: 1.51 gC m$^{-2}$ year$^{-1}$/IQR: [0.44–5.41]), but might reach similar values in specific ecosystems (e.g., 124 gC m$^{-2}$ year$^{-1}$ of aquatic insects flowing from lakes to tundras in Iceland). Lastly, few studies document spatial flows of vertebrate origin (3.1%), and those which do report highly contrasting values: feces of foraging deer or fishery discards represent small flows of around 0.1–1 gC m$^{-2}$ year$^{-1}$, while the net inputs to freshwater ecosystems of hippopotamus defecating into rivers, or of drowned migrating wildebeests, range between 100 and 1000 gC year$^{-1}$ per meter squared of river. Overall, cataloging the types of spatial flows documented so far, and their drivers (Fig. 2), also reveals that a large portion of the quantified spatial flows concerns the terrestrial-freshwater (66.9%) or pelagic-benthic (18.0%) interfaces, and are driven by passive processes (70.5%) rather than active transports by animals.

**Comparison of cross-ecosystem flow to local flux magnitudes**. To assess the potential importance of these spatial flows to ecosystem functioning, we compared their magnitude to those of local biological fluxes within each receiving ecosystem. This comparison is conservative regarding the importance of spatial flows: local and spatial fluxes may not be independent, because spatial flows likely enhance local fluxes in recipient ecosystems. Magnitudes of spatial flows versus local fluxes are similar in freshwater and in some benthic systems, whereas spatial flows are generally two to three orders of magnitude smaller than local fluxes in terrestrial ecosystems (Fig. 3). These patterns result from freshwater and benthic ecosystems displaying low gross primary production (GPP) while receiving abundant cross-ecosystem inflows, which contrasts with terrestrial ecosystems receiving little

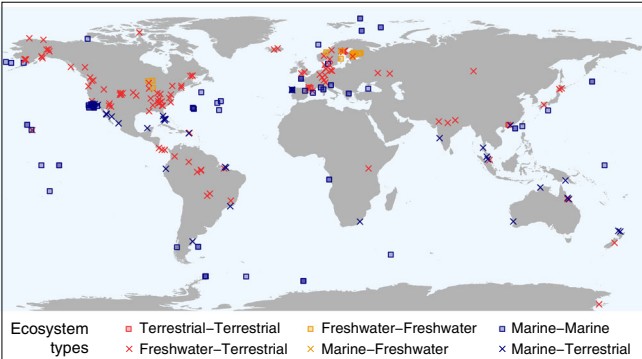

**Fig. 1** Global distribution of quantifications of cross-ecosystem flows of carbon. Colors and shapes indicate the type of ecosystems coupled by cross-ecosystem flows of carbon: terrestrial (i.e., forest, grassland, agro-ecosystem, desert), marine (i.e., ocean pelagic, ocean benthic) and freshwater (i.e., stream, lake, wetland). The map is made with Natural Earth. Free vector and raster map data @ naturalearthdata.com

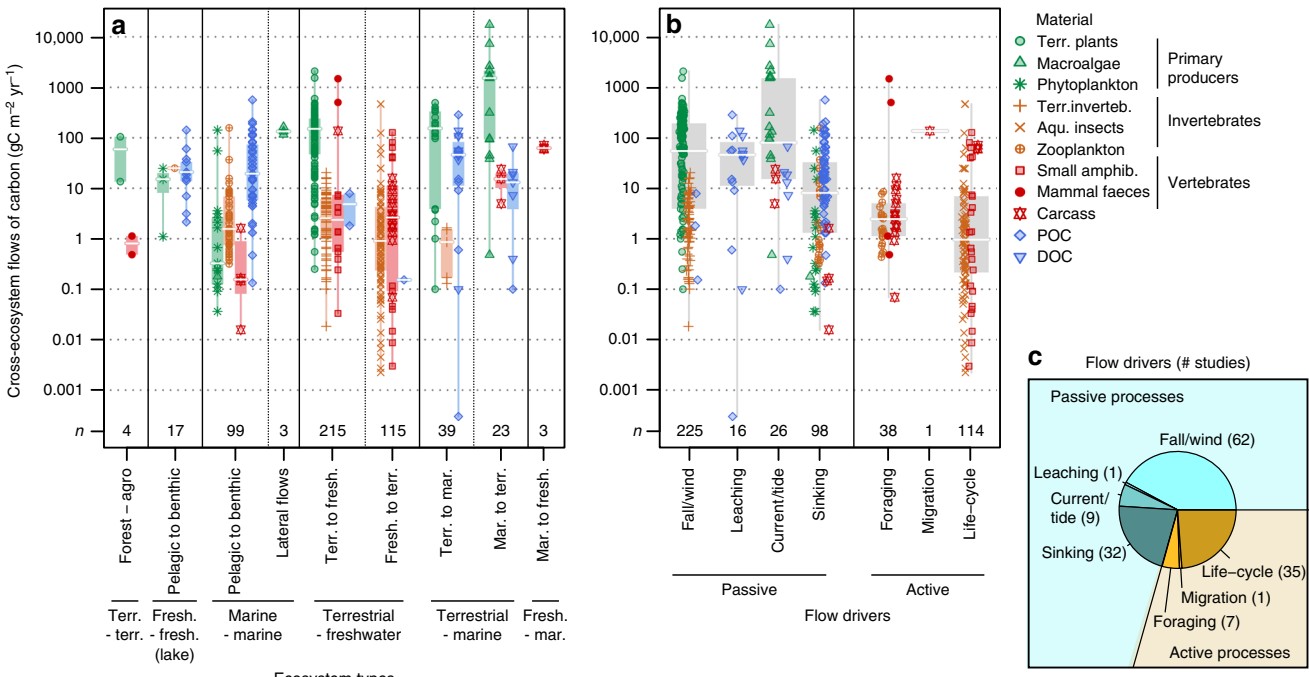

**Fig. 2** Cross-ecosystem carbon flows. Values are provided **a** by types of ecosystems connected by the flows, with vertical labels specifying ecosystems (agro stands for agro-ecosystem) and flow direction, and **b** by underlying flow drivers. Values are in gC m$^{-2}$ y$^{-1}$ on a log scale and $n$ indicates the number of data points. The areal unit, m$^2$, refers to the recipient ecosystem. Boxplots give median (white line), 25 and 75% percentiles (box), and range (whiskers). **c** Number of studies per driver type

inflows but producing abundant biomass (e.g., median GPP of 55.2 vs. 611.5 gC m$^{-2}$ year$^{-1}$ and median spatial inflows of 59.9 vs. 1.3 gC m$^{-2}$ year$^{-1}$ for streams and grasslands, respectively). In addition, ecosystem respiration in freshwater and benthic systems often exceeds local primary production, as indicated by average negative net ecosystem productions (NEP) (Fig. 4; for instance: median/[IQR] for streams: −114.1 gC m$^{-2}$ year$^{-1}$ / [−400.5 to −20.9], and marine benthic ecosystems: −4.1 gC m$^{-2}$ year$^{-1}$/ [−103.7–39.5]), despite noticeable variability. This could be due, for example, to differences in light availability in shallow tropical sea grass meadows (more positive NEP values) vs. deep or turbid waters (more negative NEP values) promoting or constraining photosynthesis respectively. By contrast, terrestrial and pelagic ecosystems have on average a statistically significant net autotrophic functioning, with confidence intervals of mean NEP all lying above zero (see Fig. 4 and Supplementary Table 1; median/ [IQR] for pooled terrestrial ecosystems: 129.0 gC m$^{-2}$ year$^{-1}$ [20.0–293.0], and pelagic ecosystems: 40.0 gC m$^{-2}$ year$^{-1}$/ [17.5–86.9]; overall, they have an 80% probability of being autotrophic based on our dataset), and receive negligible spatial flows compared to their local production (e.g., IQR of spatial inflows versus GPP in forests is [0.16–6.70] vs. [1064–1786] in gC m$^{-2}$ year$^{-1}$), except in deserts (see Fig. 3).

## Discussion

Overall, our extensive carbon-based synthesis of cross-ecosystem flows characterizes strong, and relatively unidirectional, spatial couplings in which freshwater and unproductive benthic eco-systems receive quantitatively important material exported from terrestrial and pelagic ecosystems, respectively (Fig. 5). We thus provide a quantitative confirmation of the long-held belief that things flow down hills[25,26] and therefore have the potential to propagate changes from higher to lower elevation- communities[27]. While the existence of strong spatial flows reported here

does not necessarily imply that these potential resources are integrated into recipient food webs—recalcitrant material could merely accumulate—it is indeed common that more carbon is respired than could be locally produced in freshwater and benthic systems (Fig. 4, and values in previous section), indicating that ecosystem function must rely on these allochthonous resources. Naturally, this dependency might vary with individual ecosystems' degree of heterotrophy. Most of streams and lakes are net heterotrophic (around 75% probability based on our dataset; see Fig. 4), but some can be net autotrophic when conditions allow greater primary production (e.g., lower riparian cover). Similarly, benthic marine systems include some of the most productive ecosystems, such as sea grass beds or coral reefs in shallow tropical lagoons, for which the contribution of spatial flows to total production might be negligible. However, the decrease of photosynthesis with depth or water turbidity shifts benthic ecosystems toward a detritivore-based functioning relying essentially on spatial exports from more autotrophic systems[28]. In addition, an increasing number of isotopic studies and studies manipulating the magnitude of flows demonstrate the role of spatial flows in subsidizing and structuring recipient communities[22,23]. For instance, terrestrial litter exclusion can alter entire stream food webs through cascading effects from detritivores to predators[29]. Furthermore, the degree of integration into recipient food webs can depend on the characteristics of the carbon introduced there, with, for instance, terrestrial dissolved vs. particulate organic carbon, respectively, subsidizing pelagic bacteria versus benthic invertebrates and zooplankton in lakes[30]. Such dependency on spatial resources implies that unproductive freshwater and benthic systems are especially sensitive to alterations of spatial carbon flows and, thus, to changes in donor ecosystem dynamics[31–33].

In other ecosystems—such as most terrestrial ones—spatial carbon flows seem quantitatively negligible compared to local

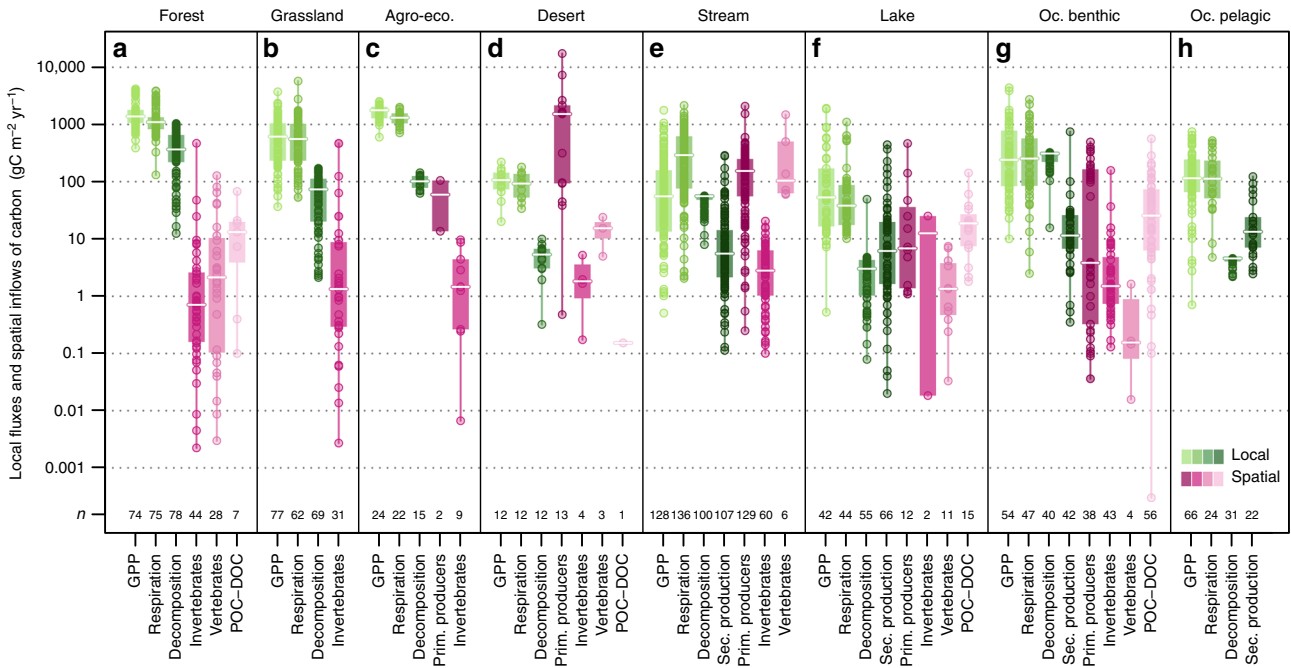

**Fig. 3** Comparison of local fluxes versus cross-ecosystem spatial inflows of carbon. Local fluxes within (green), and cross-ecosystem flows to (pink), specific ecosystem types (**a–h**). Panel titles give the ecosystem type (Oc. = Ocean). Local fluxes depicted are gross primary production (GPP), ecosystem respiration, decomposition flux, and secondary production (in aquatic ecosystems only). Cross-ecosystem flows are clumped into imported material of four origins: primary producers, invertebrates, vertebrates, and organic carbon (particulate and dissolved: POC-DOC). No spatial flow of carbon to pelagic ocean ecosystems with areal units was found. Circles give values in gC m$^{-2}$ year$^{-1}$ on a log scale, and $n$ the number of data points. Boxplots give median (white line), 25 and 75% percentiles (box), and range (whiskers). Whiskers and three null values of GPP in streams are omitted due to log scale. Note that fluxes of GPP, measured locally, may not be independent from cross-ecosystem flows, which might contribute to local production

fluxes (Fig. 3; note: only few studies are reporting flows between terrestrial ecosystems, indicating a potentially important knowledge gap). However, the low magnitude of these average flows to terrestrial systems does not exclude high local impacts when they are constricted to small areas. For example, material deposition from freshwater or marine ecosystems is locally high on shorelines but rapidly decreases with distance from source[34]. Moreover, even some terrestrial ecosystems can be net heterotrophic (around 20%), for instance due to perturbations like fires, harvesting, or drought, which could disrupt production capacity. In deserts, where local production is very limited (IQR of GPP: [76.6–138.5] in gC m$^{-2}$ year$^{-1}$, compared to in average 1379.5 gC m$^{-2}$ year$^{-1}$ in forests), materials provided by other less limited ecosystems (e.g., oceans) can be substantial (IQR of spatial flows to deserts: [4.95–1575.3] in gC m$^{-2}$ year$^{-1}$), and represent essential subsidies for detritivore-based communities[35]. Overall, in terrestrial and pelagic ecosystems, carbon is generally not a limiting element and whole-ecosystem production should be mostly driven by local primary production rather than by allochthonous carbon flows.

However, carbon is intimately linked to nutrients within biological molecules. Though quantitatively small in terms of carbon, spatial flows to terrestrial systems often have higher nitrogen content than the vegetation-based subsidies these systems export (e.g., insects or salmon vs. leaves in terrestrial-freshwater couplings[22,34,36,37]), and this has been shown to structure terrestrial communities[20,22]. Some cross-ecosystem flows, not considered here due to our focus on carbon units, are even predominantly of nitrogen or phosphorus, enriching terrestrial (e.g., the well-studied salmon-to-forest subsidy[38], or guano from foraging sea birds[39]) or pelagic systems (e.g., excretion of marine mammals[40]). Carbon and nutrients in cross-ecosystem material

flows may relax different limitations in recipient ecosystems, thus enhancing regional production[13,22]. Such interdependencies linking the productions of different ecosystems might affect carbon fixation at larger, meta-ecosystem scales, which we should account for in our general effort to regulate the global carbon cycle via land use management.

Our synthesis also shows important gaps in the quantification of spatial couplings. Passive flows of detritus or of small organisms are relatively well documented, as can be easily measured by various traps. By contrast, it is more challenging to quantify active spatial flows resulting from animal movement. This requires indirect estimations, for instance combining animal tracking and measurements of ingestion and excretion rates[41,42]. Such quantifications are rare (Fig. 2), but the few existing studies suggest that widespread animal movements, such as foraging or migration, could act as important ecosystem connectors[43,44]. In particular, mobile species using multiple ecosystems[45], or those whose behavior leads to massive aggregations of individuals, likely move important amounts of material among ecosystems. At the moment, the lack of data hinders an assessment of the global importance of these actively-driven cross-ecosystem subsidies[46]. However, such linkages could render common ecosystem couplings more bi-directional than previously thought, as proposed for the aquatic-terrestrial interface[34,47]. Detailed evaluation of flow bi-directionality is essential to better understand spatial feedbacks between connected ecosystems. Resource exchanges between ecosystems underlie both optimization in resource use at the landscape scale, and potentials for spatial amplification of local perturbations. An examination of the ultimate fate of spatial flows in recipient ecosystems was beyond the scope of our dataset, but next challenges for spatial ecology must include such assessments, notably considering quantity versus quality effects of

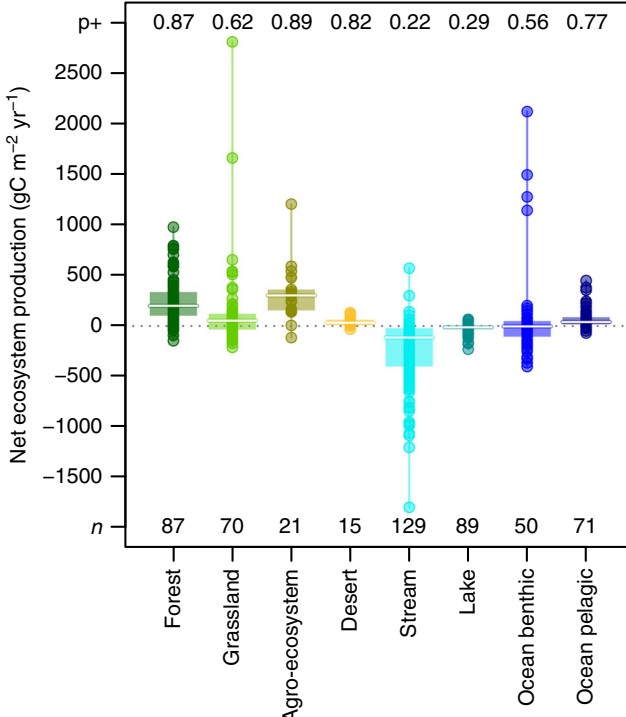

**Fig. 4** Net ecosystem production for different ecosystem types. Net ecosystem production (NEP) corresponds to the balance between gross primary production and ecosystem respiration. Negative values denote net heterotrophic functioning. Circles give individual values in gC m$^{-2}$ year$^{-1}$. Note that these are production fluxes and not productivity rates (biomass turnover), for which we would have higher values in aquatic compared to terrestrial ecosystems. Boxplots give median (white line), 25 and 75% percentiles (box), and range (whiskers). Top numbers (p+) give the probability of NEP being positive within each ecosystem type, assuming normal distributions of the data (quantile corresponding to 0). Bottom numbers (n) indicate the number of data points. The 95% confidence intervals for mean NEP within each ecosystem types are in gC m$^{-2}$ year$^{-1}$: Forest [204; 301], Grassland [28; 224], Agro-ecosystem [197; 438], Desert [18; 70], Stream [−307; −193], Lake [−32; −14], Ocean benthic [−60; 201], and Ocean pelagic [47; 90] (see full results of two-sided t-test in Supplementary Table 1). A conservative non-parametric Kruskal-Wallis' test indicates significant differences among ecosystems ($\chi^2_{7,532} = 275.04, P < 0.001$). The post-hoc multiple comparisons test using rank sums gives the following groups: a, bc, a, abcd, e, d, bd, ac (lower case letters for grouping from left to right in the figure; see Methods)

spatial flows, as well as relating animal activities at different scales to fluxes of resources linking ecosystems[46].

Overall, our global analysis highlights the ubiquity of cross-ecosystem spatial flows of carbon, and the variety in their magnitude. The importance of such flows for ecosystem functioning is variable among ecosystem types, but general knowledge of an ecosystem's limiting elements and internal metabolism (net heterotroph vs. autotroph) enables some reliable predictions: notably, freshwater and benthic ecosystems might be especially vulnerable to alterations of spatial carbon flows, and by extension to perturbations in the autotrophic ecosystems from which these flows arise. However, gaps in flow quantification leave some uncertainty as to whether different ecosystems might act as buffers or amplifiers of spatial dynamics within landscapes. Documenting these ecological blind spots is necessary to improve our ability to predict ecosystem responses to global changes across landscapes.

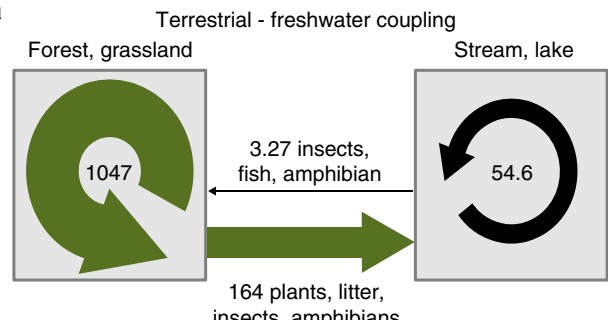

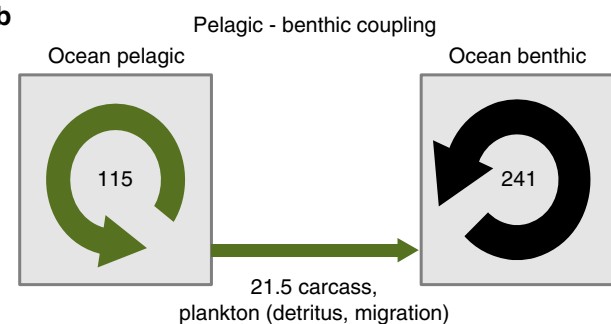

**Fig. 5** Well-documented natural meta-ecosystems. **a** Terrestrial–freshwater and **b** pelagic–benthic ecosystem spatial couplings. Round and horizontal arrows represent gross primary production (GPP) and cross-ecosystem flows, respectively. Numbers are median values for GPP and sum of median values of cross-ecosystem flows of each different origin (primary producer, invertebrate, vertebrate, and particulate and dissolved organic carbon), both expressed in gC m$^{-2}$ year$^{-1}$. Width of arrows is proportional to these values based on double squared root transformation. Inter-quartile ranges [IQR] for GPP: terrestrial [589–1497], freshwater [15.3–160.5], ocean pelagic [66–243], ocean benthic [83–745]; cross ecosystem flows (sum of IQR of flows of different origins): to terrestrial [0.38–15.09], to freshwater [52–272], to ocean benthic [6.9–79.9]

## Methods
**Principle of the study**. We conducted an extensive literature review of empirical values of cross-ecosystem spatial flows of carbon over the globe (distribution in Fig. 1), and compared their magnitude to local fluxes within ecosystems receiving these flows. We chose carbon as the focal material unit to profit from widely available carbon data for local fluxes. To enable the spatial flow to local flux comparison, we only considered measurements of spatial flows that were either provided in or could be converted into gC m$^{-2}$ year$^{-1}$.

**Data collection**. Our systematic search covered four broad categories of terrestrial ecosystems (forest, grassland, agro-ecosystem, and desert) and four of aquatic ecosystems (stream, lake, pelagic ocean, and benthic ocean). We considered all ecosystems (if available) in five major global climatic zones (arctic/alpine, boreal, temperate, tropical, and arid). Supplementary Table 2 provides the definitions of ecosystem categories and climatic zones. For marine ecosystems, we grouped arctic, boreal, temperate vs. arid and tropical climates into Cold and Warm respectively, to account for a lesser influence of climate on oceanic systems due to the buffering effect of large water volumes. For each relevant ecosystem x climatic zone combination (see Supplementary Fig. 1), we collected local carbon flux and spatial carbon flow data. We used all possible combinations of these categories and terms with similar meanings (see Supplementary Table 2) in our systematic search (see details in the next paragraphs).

We collected available values of spatial flows linking the above-mentioned different ecosystems, and which could be converted into gC m$^{-2}$ year$^{-1}$ in order to homogenize data and make comparisons possible. The latter constraint excluded cross-ecosystem flows of nutrients for which no carbon equivalent was possible, and flows expressed without information of the area of influence in the ecosystem receiving the flow. For the first case, this primarily excluded studies based purely on isotopic methods, for example those reporting the proportion of marine-derived nutrients in riparian plants, and those reporting transport of nutrients without information on nutrient to carbon ratios. Thus, several well-studied spatial flows such as those triggered by salmon moving from marine to freshwater ecosystems, terrestrial fertilization by seabird guano, and nutrient excretions by fish or marine

mammals, are not well represented in our dataset. For the second case, measurements of amount of dissolved organic carbon or sediments flowing from streams into estuaries[48] where excluded because the area of the recipient ecosystem was undefined. Terms primarily used for the search of spatial flows were "(subsid* OR spatial flow*) AND ecosystem", with "ecosystem" also being replaced by specific ecosystems or pairwise combinations of ecosystem and climate types of interest. For example, to search for spatial flows from streams to grasslands, we used the search strings "(subsid* OR spatial flow*) AND (stream OR river OR aquatic) AND (grassland OR prairie OR meadow)", replacing the last term by "(tundra OR grassland OR meadow) AND (arctic OR alpine)" when we looked more precisely at stream subsidies to grasslands in cold climates.

In addition, for each ecosystem x climatic zone combination, we systematically searched published literature for values of the following within-ecosystem carbon fluxes: gross primary production (GPP), secondary production (in aquatic ecosystems only), ecosystem respiration ($R_e$), net ecosystem production (NEP), and decomposition fluxes. We chose GPP rather than NPP (net primary production) to allow a more straightforward comparison with the spatial flows entering a system, which are also gross fluxes. Since decomposition fluxes were rarely directly provided, we derived them from detritus stocks and decomposition rates (see next section for calculations). A first systematic search was conducted by using all possible combinations of the names of each ecosystem type, climatic zone and flux of interest, with small variation when relevant (e.g., "decomposition OR decay" for decomposition flux and rates). The different terminologies used across various research fields to describe the same processes, and the fact that the data of interest were often located in different sections of the studies (Methods vs. Results) limited the efficiency of standardized keyword search across the data types. We therefore complemented the dataset with multiple customized searches until we compiled a minimum number of ten independent values of each variable of interest (i.e., different fluxes, detritus stock, and decomposition rate) for each ecosystem x climatic zone combination. At the end, data were pooled by ecosystem type.

In total, we collected 3034 values from 557 published studies, including 518 values of cross-ecosystem subsidies. A summary of all values and the respective references are provided in Supplementary Table 3 (cross-ecosystem flows) and Supplementary Table 4 (local fluxes).

**Calculations used for data extraction.** When only two of three major fluxes (gross primary production, ecosystem respiration, and net ecosystem production (GPP, $R_e$, and NEP, respectively) were reported, we estimated the unreported flux:

$$NEP = GPP - R_e \tag{1}$$

$$NEP = NPP - R_h \tag{2}$$

$$NPP = GPP - R_a \tag{3}$$

NPP is the net primary production, $R_h$ the heterotrophic respiration and $R_a$ the autotrophic respiration.

We derived decomposition fluxes $D_F$ from detritus stocks $D_M$ and decomposition rates $k$, with the classical exponential decay model:

$$D_F = D_M\left(1 - e^{-kt}\right) \tag{4}$$

To calculate individual decomposition flux values, we parameterized detritus stock $D_M$ with the median values of all detritus stocks in a given ecosystem x climatic zone combination and used the decomposition rate values collected from the literature to produce flux values. The rates were values of $k$, the first order constant in the classical exponential decay model. When not directly provided, we derived $k$ with one of the equations proposed by Cebrian and Lartigue[6] depending on the data available in the study:

$$D_t = D_{t_0}e^{-k(t-t_0)} \tag{5}$$

$$D_F = (D_P - E)\left(1 - e^{-kt}\right) \tag{6}$$

In Eq. 5, $D_t$ is the detrital mass at time $t$ and $D_{t_0}$ the initial detrital mass. This equation was used when decomposition was estimated as the proportion of detrital mass loss $\left(1 - D_t/D_{t_0}\right)$ via a litter-bag experiment, a classical method in freshwater and terrestrial ecology. In Eq. 6, $D_F$ is the (absolute) decomposition flux during the study period $t$, that is the flux from detritus stock to bacteria and other detritivores, $D_P$ is the detritus production, and $E$ the detritus export (e.g., sedimentation). In few cases of ocean pelagic data, we used the microbial loop of primary production vs. bacterial production to parameterize $D_P$ and $D_F$, respectively. If not available, the export rate was set to 0, leading to $k$ underestimation, which is conservative in our cross-ecosystem comparison given that $k$ is already at the higher end of the range in these pelagic systems.

**Unit conversions.** Once collected, we standardized values by converting them all into areal carbon units, that is, gC m$^{-2}$ for detritus stocks and gC m$^{-2}$ year$^{-1}$ for local fluxes and cross-ecosystem subsidies. Decomposition rates were expressed in year$^{-1}$.

Carbon conversion: We used data in carbon units (gC) when it was directly provided in the study, or we calculated the values using carbon content when reported in the study (50% of data points). Alternatively, we applied the most specific factor to convert the data into carbon units depending on the level of detail available on the material of interest (see Supplementary Table 5 for conversion factors). For decomposition rates, we did not transform units into carbon. We made the most parsimonious assumption that carbon loss rate is identical to loss rate in the unit provided (generally dry weight or ash-free dry weight). While this is a simplification, we concluded that this best allowed us to keep measurements consistent across data sources, in the absence of more detailed information.

Time extrapolation: 55% of local fluxes or rates were already provided in yearly units. For the others, we extrapolated to the year by using the number of days in the growing season as reported in the study, or the ice-free period in cold climates. When growing season length (GSL) was not specified in the study we used averaged estimates detailed by Garonna et al.[49] for the different climatic zones in Europe[50]: 181 days for temperate climate (mean of atlantic and continental), 155 days for boreal, 116 days for arctic, and 163 days for arid systems (mean of Mediterranean and steppic). We assumed no strong seasonality in tropical climates (365 days of GSL). We did not apply any conversion if the value was measured on a study period longer than the above GSL for the corresponding climate.

Volume to area conversions and depth integration: Some data were given per unit of volume. For freshwater systems, we converted the data into area units by integrating them over the water column, using the mean depth of the river or lake. When not directly available in the study we calculated depth by dividing the volume per the area in lakes, or by estimating depth from discharge in rivers with the formula depth $= c \times Q^f$, with $c = 0.2$, $f = 0.4$ and $Q$ the discharge in m$^3$ s$^{-1}$ (see ref. [51]). For small catchment areas, that is <1 km$^2$, we estimated the depth to be 5 cm based on known river scaling-properties[51]. For marine data, notably production in the pelagic zone, studies generally provide a meaningful depth, which defines the euphotic zone such as the Secchi depth or the 1% light inflow depth. We integrated values in volume units over this depth, and to 100 m depth when only sampling depths were provided.

Areal units for cross-ecosystem flows: Since we were interested in comparing the magnitude of spatial flows to that of the recipient ecosystem's own local fluxes, we needed cross-ecosystem flows measured in areal units of the recipient ecosystem. Depending on the method of quantification, cross-ecosystem flow can be directly expressed in this way (e.g. litter traps in the recipient ecosystem). In other cases, for example with vertical fluxes between pelagic and benthic ecosystems, the equivalence of donor and recipient ecosystem areas affected by the flow is obvious. However, in the case of lateral flows occurring from aquatic to terrestrial ecosystems (e.g., emergent insects, carcasses of salmon caught by bears), flows were often provided per m$^2$ of donor area (21% of spatial flows). We could not use such measurements directly because the magnitude of the flow in the recipient ecosystem depends on both the total surface of production and the boundary length. For instance, lakes with the same area but having circular vs. complex-shorelines will lead, for the same total emergent insect flux, to higher versus lower magnitudes of flows, respectively, distributed per areal unit of the recipient ecosystem. Moreover, the maximum influence of such cross-ecosystem flow is found near at the shoreline and decreases with distance from the shore (for example, in aquatic insects[19]). To adjust for this in a conservative approach, we homogenized our aquatic-to-terrestrial flows assuming a uniform distribution of the flow on the first ten meters from the shore, a distance within which most of the aquatic insect flows fall[52] (but could fall substantially farther in some specific systems[53]), or most of the salmon carcasses brought by bears on land are deposited[54]. Therefore, when the values of spatial flows were provided in areal unit of the donor ecosystem (aquatic), we first calculated the flow per meter shoreline (in some cases, values were already provided per meter shoreline; e.g., wrack deposited on beaches), and then divided this value by 10 m of influenced land. To calculate the spatial flow per shoreline length in freshwater systems, we followed the method described by Gratton and Vander Zanden[19] for insect emergence data: In lakes we multiplied the data per the total lake area and divided per the perimeter. When not directly available, the perimeter was approximated by $2 \times D_L \times$ (area $\times \pi)^{0.5}$, with $D_L$ the development factor defined by Kalff[55], which is 1 for circular shapes, 2 for the same area with a two-fold larger perimeter. In streams we multiplied the original donor area flow value by stream width, and then divided it per two (riversides) to obtain the flow per shoreline length. Supplementary Table 6 shows that our conclusions are not sensitive to the choice of the distance from shoreline used to calculate the recipient terrestrial area of aquatic subsidies: using a more conservative threshold of 100 m only decreases the importance of carbon spatial flows to terrestrial ecosystems, that we already assess as low compared to terrestrial local fluxes.

**Flow drivers.** We defined categories of flow drivers to examine the underlying processes of documented cross-ecosystem subsidies. Drivers could be either passive, via physical processes such as gravity, wind, water currents, tides, or diffusion, or active, via animal movements such as those triggered by foraging behaviors,

seasonal migration, or cross-ecosystem movement of animals needing to complete a life-cycle. We kept these last three categories for active drivers and we clumped the passive drivers into categories reflecting broad classes of spatial flows: Fall/wind for aerial transport from terrestrial systems, Leaching for diffusion processes, Current/tides for lateral flows from aquatic systems and Sinking for vertical passive flows in the water column (see Supplementary Table 7).

**Statistical analysis of NEP**. We tested whether mean NEP of each ecosystem type was significantly different from zero with a Student's *t*-test (two-sided). Full results are reported in Supplementary Table 1. Moreover, we tested the differences of NEP among ecosystem types using the non-parametric Kruskal–Wallis' rank-test (because variances were heterogeneous according to the Bartlett's test). We then used post-hoc multiple comparison tests with the function kruskalmc of the pgirmess R package 1.6.6, based on the methods in Siegel and Castellan[56], to determine which ecosystem types were significantly different (see Fig. 4 and associated legend).

**Software**. We analyzed the data and plotted the figures with the open source software R 3.3.3[57] and the R-packages ggmap[58]2.6.1, maps[59]3.2.0, and pgirmess[60]1.6.6. Final artwork was realized with Illustrator CC 22.0.1.

**Reporting Summary**. Further information on research design is available in the Nature Research Reporting Summary linked to this article.

## Data availability
Data that support the findings of this study are summarized in Supplementary Table 3 and Supplementary Table 4, with all associated references in Supplementary Data 1. Source data files are available from the corresponding authors upon request. A reporting summary for this article is available as a Supplementary Information file.

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

## Acknowledgements

We thank Marcel Holyoak and Emanuel A. Fronhofer for discussions, and Mary O'Connor for comments on the manuscript. Funding is from the Swiss National Science Foundation Grants PP00P3_150698 and PP00P3_179089 and the University of Zurich Research Priority Program Global Change and Biodiversity (URPP GCB) to FA.

## Author contributions

I.G., C.J.L., E.H. and F.A. conceived the study and collected the data. I.G. analyzed the data, produced the figures and wrote the first draft of the manuscript. All authors edited and contributed to the final version of the manuscript.

## Additional information

**Competing interests:** The authors declare no competing interests.

