## [Peer Review File · Nature Communications]

Reviewers' comments:

Reviewer #1 (Remarks to the Author):

Review of Gounand et al.

Beyond local functioning: cross-ecosystem carbon flows connecting ecosystems.

Nature Communications NCOMMS-18-20497

This paper gathers a vast amount of data to ascertain how much fixed carbon that drives the metabolism of various ecosystems is derived from outside the system and transported to it by nature. To some extent the results are not surprising: very little in forests and a lot in stream and lake ecosystems. What is new here is the extent of the global synthesis to look for general patterns amidst a fair amount of natural variation, which is acknowledged and documented by the authors.

This paper provides a further quantitative assessment of some of the philosophical posits by Reiners and Driese (ref. 10). (Note those authors have expanded their analysis in a recent book out of Cambridge University Press). The present paper also offers some important perspectives on the arguments for net heterotrophy by the world's lakes and oceans.

Perhaps I am old-fashioned, but I would have focused, at least for the terrestrial ecosystems, on data originally gathered as net primary production (NPP), rather than the newer attempts to estimate carbon flux from eddy-covariance studies (GPP). After all, NPP is normally measured as a tangible entity—fixed organic matter as leaves, particles, or dissolved molecules, so it is thus directly comparable to the forms that are delivered from outside the system—e.g., blowing leaves, transported particles, streamwater DOC.

Figure 2 is very hard to read as it is currently drafted. This should be fixed before the paper might be published. The more seminal figure 3 is fine.

NB

Line 98: Replace “flows occurring at” by “the”

Line 138 Replace “does rely” by “relies”

Lines 301 & 309 Round these percentages to 50% and 55%, respectively

Reviewer #2 (Remarks to the Author):

Summary:

Ecosystems are open to flows of materials, organisms, and energy. Spatial ecosystem ecology has a long history of studying these flows and meta-ecosystem theory has recently emerged as a guiding framework for understanding the impacts of spatial flows among ecosystems. The authors conduct a global meta-analysis of cross-ecosystem carbon flows and relate these cross-ecosystem flows to in situ ecosystem functions (i.e., GPP, Respiration, Decomposition, Secondary production). They find that cross-ecosystem carbon flows were similar to local carbon flows in freshwater and benthic ecosystems but much lower than local carbon flows in terrestrial ecosystems. This study presents comprehensive evidence of strong ecosystem coupling and the need to consider these couplings when managing natural resources.

General comments:

1) Full disclosure: I reviewed an earlier version of this ms. I liked it the first time I read it and I still like it. The authors have addressed the elephant in the room (i.e., subsidy) and have focused the ms on reporting flux patterns. Overall, I think this is an improved contribution. Below, I revisit my original queries as I still think the ms can be improved in some aspects.

2) The breadth of this synthesis is impressive. Kudos to the authors for undertaking this adventure!

3) In most cases, the authors refer to “resource” flows but in other cases simply spatial flows. This is where determining if the flow is a subsidy or not becomes important. Language such as “resource” flows implicitly assumes the flows are subsidies (i.e., they are a resource for some recipient). But in some cases, the flows reported by the authors are likely to be “consumer” flows (e.g., some invertebrates, small amphibians, salmon). Allen & Wesner 2016 Ecology (now cited by the authors) shows a clear distinction between resource and consumer flows. All this to say that I recommend the authors remove “resources” from the language here as doing so implies the flows are all resources and all subsidies. The authors do not have the data to support this claim.

4) The distinction between cross-ecosystem and local ecosystem is critical to this analysis but unclear. Let’s take an example. If all ecosystems are open to flows, how do we know that any measure of GPP in a “local” ecosystem is independent of such flows? How many cases in the dataset do we have one ecosystem that is considered in the cross-ecosystem vs local ecosystem case. Maybe

there are long-term studies of stream X. One study reports GPP of stream X. Would this be considered a measure of local functioning? What if another study reports contribution of terrestrial invertebrates to GPP in stream X? I assume this would fall under the author's data on cross-ecosystem flows. No ecosystem in the natural world is closed and the concept of an ecosystem is scale invariant. In their comparison, the authors seem to treat local ecosystems as being closed.

5) The data show sizable variation in net ecosystem production within ecosystems and all ecosystems overlap 0. This means that within any given ecosystem category, we can have net autotrophic and net heterotrophic cases. Does the general distinction fall apart if the variation depicts such patterns? Alternatively, is the variation in net ecosystem production important to consider here?

6) Thank you for adding some details on the number of studies requiring unit conversions and addressing some of the assumptions of your conversion factors in this revised ms.

Specific comments:

7) line 107-113. The wording here is awkward – consider rephrasing.

8) line 117 and 120. Remove “a” in both cases.

9) lines 129-132. This is not new but the novelty lies in the huge scope of the data. I recommend the authors re-phrase this along the lines of “Our extensive synthesis of X confirms the long-held belief that ‘things flow down hills’ (i.e. Lindeman, Leroux)...” Phrasing it this way does not take away from the value of the work.

10) lines 132-134. I appreciate this new sentence but I think it would be better placed near the end in a section highlighting future directions. I think the subsidy point is made very well with lines 134-139.

11) line 138. Replace “does” with “must”.

12) lines 141-144. Nice example but consider adding specific details (study system, global synthesis, etc.). This paper has an impressive amount of data yet it is still very conceptual. Real examples with specific details will help ground it for the readers.

Reviewers' comments:

Reviewer #1 (Remarks to the Author):

Review of Gounand et al.

Beyond local functioning: cross-ecosystem carbon flows connecting ecosystems. Nature Communications NCOMMS-18-20497

This paper gathers a vast amount of data to ascertain how much fixed carbon that drives the metabolism of various ecosystems is derived from outside the system and transported to it by nature. To some extent the results are not surprising: very little in forests and a lot in stream and lake ecosystems. What is new here is the extent of the global synthesis to look for general patterns amidst a fair amount of natural variation, which is acknowledged and documented by the authors.

This paper provides a further quantitative assessment of some of the philosophical posits by Reiners and Driese (ref. 10). (Note those authors have expanded their analysis in a recent book out of Cambridge University Press). The present paper also offers some important perspectives on the arguments for net heterotrophy by the world's lakes and oceans.

Authors' response: We thank the reviewer for these positive and constructive comments. We have added the reference of Reiners and Driese's book, and integrated all subsequent suggestions as well.

Perhaps I am old-fashioned, but I would have focused, at least for the terrestrial ecosystems, on data originally gathered as net primary production (NPP), rather than the newer attempts to estimate carbon flux from eddy-covariance studies (GPP). After all, NPP is normally measured as a tangible entity—fixed organic matter as leaves, particles, or dissolved molecules, so it is thus directly comparable to the forms that are delivered from outside the system—e.g., blowing leaves, transported particles, streamwater DOC.

Authors' response: We agree that multiple measures have been used to quantify primary production, and that NPP is widely used in terrestrial systems. However, we specifically chose GPP to compare it with spatial flows because both are gross fluxes (in the sense that it is what actually flows), and thus more equivalent. Moreover, the homogenization of units into $\text{gC m}^{-2} \text{y}^{-1}$ allows to compare fluxes measured with different methods, and of different material (CO₂ fixed versus more tangible entities transported, such as leaves); thus, we took care of maintaining consistency when comparing GPP with spatial flows. We have now integrated additional text in the method to justify our choice, lines 273-274:

“We chose GPP rather than NPP (Net Primary Production) to allow a more straightforward comparison with the spatial flows entering a system, which are also gross fluxes.”

Figure 2 is very hard to read as it is currently drafted. This should be fixed before the paper might be published. The more seminal figure 3 is fine.

Authors' response: Thank you for this feedback. We have now rearranged the presentation of figure 2 to make it more readable, notably by putting panel c into a rectangle and illustrating some of the flows. We think this makes it indeed more readable and clearer. We are happy to adapt it further if specific suggestions about possible changes are given.

NB

Line 98: Replace "flows occurring at" by "the"
Authors' response: done accordingly.

Line 138 Replace "does rely" by "relies"
Authors' response: replaced by "must rely" as suggested by reviewer 2.

Lines 301 & 309 Round these percentages to 50% and 55%, respectively
Authors' response: done accordingly.

Thank you very much for your constructive feedback on our manuscript.

Reviewer #2 (Remarks to the Author):

Summary:

Ecosystems are open to flows of materials, organisms, and energy. Spatial ecosystem ecology has a long history of studying these flows and meta-ecosystem theory has recently emerged as a guiding framework for understanding the impacts of spatial flows among ecosystems. The authors conduct a global meta-analysis of cross-ecosystem carbon flows and relate these cross-ecosystem flows to in situ ecosystem functions (i.e., GPP, Respiration, Decomposition, Secondary production). They find that cross-ecosystem carbon flows were similar to local carbon flows in freshwater and benthic ecosystems but much lower than local carbon flows in terrestrial ecosystems. This study presents comprehensive evidence of strong ecosystem coupling and the need to consider these couplings when managing natural resources.

General comments:

1) Full disclosure: I reviewed an earlier version of this ms. I liked it the first time I read it and I still like it. The authors have addressed the elephant in the room (i.e., subsidy) and have focused the ms on reporting flux patterns. Overall, I think this is an improved contribution. Below, I revisit my original queries as I still think the ms can be improved in some aspects.

Authors' response: We thank the reviewer for the very helpful comments provided on this, but also on the previous, manuscript version. As noted, we have incorporated all these suggestions (e.g., regarding "subsidies"), and are happy to get such a positive feedback on our improvements as well as the overall manuscript.

2) The breadth of this synthesis is impressive. Kudos to the authors for undertaking this adventure!

Authors' response: Thanks again!

3) In most cases, the authors refer to "resource" flows but in other cases simply spatial flows. This is where determining if the flow is a subsidy or not becomes important. Language such as "resource" flows implicitly assumes the flows are subsidies (i.e., they are a resource for some recipient). But in some cases, the flows reported by the authors are likely to be "consumer" flows (e.g., some invertebrates, small amphibians, salmon). Allen & Wesner 2016 Ecology (now cited by the authors) shows a clear distinction between resource and consumer flows. All this to say that I recommend the authors remove "resources" from the language here as doing so implies the flows are all resources and all subsidies. The authors do not have the data to support this claim.

Authors' response: We completely agree with you. We do not want to imply that all flows automatically represent subsidies. We followed your suggestion and removed the word "resource" everywhere, except when we discuss subsidies explicitly; there we use the more cautious formulation "potential resource".

Based on your comment, we also double-checked and ensured that our statements are clear, such that the flows we are presenting are distinct in their nature from the dispersal flows considered in all the meta-community research. Actually, even if we often do not have the information about the ultimate fate of the spatial flows, they are either non-living material (e.g., detritus, feces), or animals dying in the recipient ecosystem (finishing their life-cycle such as with salmon or invertebrates), or considered as potential prey in the original study (such as with salamanders). In these cases, we can consider the animal movement to be the conveyer of material among ecosystems. We thus chose to call flows "material flow" or "carbon flow" in places where it could otherwise be unclear for the reader that these are not dispersal flows.

We are very thankful for your comment on avoiding misunderstanding with respect of the fate of the flows in the recipient ecosystem, and have now ensured that our terminology is consistent and clear throughout the manuscript.

4) The distinction between cross-ecosystem and local ecosystem is critical to this analysis but unclear. Let's take an example. If all ecosystems are open to flows, how do we know that any measure of GPP in a "local" ecosystem is independent of such flows? How many cases in the dataset do we have one ecosystem that is considered in the cross-ecosystem vs local ecosystem case. Maybe there are long-term studies of stream X. One study reports GPP of stream X. Would this be considered a measure of local functioning? What if another study reports contribution of terrestrial invertebrates to GPP in stream X? I assume this would fall under the author's data on cross-ecosystem flows. No ecosystem in the natural world is closed and the concept of an ecosystem is scale invariant. In their comparison, the authors seem to treat local ecosystems as being closed.

Authors' response: This is a good point. We agree that local and spatial flows may not always be independent, and we actually do not want to suggest such independence. In fact, we very much agree that spatial flows contribute to the measured GPP in recipient ecosystems, and there is an inherent entangling and feedback of local and spatial flows (which is one of the core aspects of meta-ecosystems).

However, in terms of quantifying measurements and making comparisons, we have to simply assume them as two discrete steps, which makes our conclusions conservative: the GPP locally measured (including the possible contribution of spatial resource) is necessarily larger than a GPP that would have been independent from spatial flow contribution. Therefore when spatial flows are of similar magnitude than 'non-independent GPP' magnitude, they are likely to be even more significant than if we would have had any 'independent GPP' available (if it were possible to measure such value from observational data; such data are, at least to our knowledge, not available).

Overall, we are thus likely underestimating rather than overestimating the role of spatial flows.

To make clear that our argumentation and our comparison does not assume independence of spatial and local fluxes, we have added a clarifying sentence in the legend of Figure 3, lines 620-622: “*Note that fluxes of GPP, measured locally, may not be independent from cross-ecosystem flows, which might contribute to local production.*” Moreover, we state this explicitly in the results section, lines 107-109: “*This comparison is conservative regarding the importance of spatial flows: local and spatial fluxes may not be independent, because spatial flows likely enhance local fluxes in recipient ecosystems.*”

5) The data show sizable variation in net ecosystem production within ecosystems and all ecosystems overlap 0. This means that within any given ecosystem category, we can have net autotrophic and net heterotrophic cases. Does the general distinction fall apart if the variation depicts such patterns? Alternatively, is the variation in net ecosystem production important to consider here?

Authors’ response: We agree that the variation in NEP is important and that each category of ecosystem has examples of net autotrophic and net heterotrophic cases, with different causes of this variation among ecosystem types. Importantly, however, we wanted to highlight overall and general trends (e.g., terrestrial systems are generally net autotrophic, while freshwater are not), without pretending that every individual case necessarily follows these general trends.

Based on your and the editor’s comments we further elaborate this point and now statistically support all our interpretation of general trends in the importance of spatial flows for recipient ecosystems. We also discuss the extent to which this holds within an ecosystem category.

Specifically, we have addressed this point in three ways:

First, we have added extensive statistical tests to compare among-ecosystem mean differences of NEP, and give statistical support for all our interpretation of the average trends on net autotrophy / heterotrophy of ecosystem types:

- We tested mean NEP of each ecosystem type against 0 with two-sided Student’s t-test (results in Online Supplementary Table 1) and provide the confidence intervals of means in the legend of the figure, lines 632-636: “*The 95% confidence intervals for mean NEP within each ecosystem types are in $\text{gC m}^{-2} \text{y}^{-1}$: Forest [204; 301], Grassland [28; 224], Agro-ecosystem [197; 438], Desert [18; 70], Stream [-307; -193], Lake [-32; -14], Ocean benthic [-60; 201], and Ocean pelagic [47; 90] (see full results of two-sided t-test in Supplementary Table 1).*”. This shows that terrestrial and pelagic ecosystems are on average net autotrophic while freshwater ecosystems are on average net heterotrophic, and benthic ecosystems might be one or the other.
- We also provide, in the figure, the probability of NEP being positive for each ecosystem type (to which quantile does zero correspond assuming a Gaussian

distribution) to show the proportion of each ecosystem that might differ from the average trend.

- We compare among-ecosystem differences by a statistical test as suggested by the editorial team (Kruskal-Wallis non parametric test + post-hoc multiple mean comparisons), adding a specific paragraph in the methods, lines 403-410.

Secondly, we further discuss the variability of NEP in the text, the sources of this variability, and how this could relate to variation in spatial flow importance for recipient ecosystems:

- Lines 128-129: *“overall, they [terrestrial and pelagic ecosystems] have an 80% probability of being autotrophic based on our dataset”*.
- Lines 146-155: *“Naturally, this dependency might vary with individual ecosystems’ degree of heterotrophy. Most of streams and lakes are net heterotrophic (around 75% probability based on our dataset; see Fig. 4), but some can be net autotrophic when conditions allow greater primary production (e.g., lower riparian cover). Similarly, benthic marine systems include some of the most productive ecosystems, such as sea grass beds or coral reefs in shallow tropical lagoons, for which the contribution of spatial flows to total production might be negligible. However, the decrease of photosynthesis with depth or water turbidity shifts benthic ecosystems toward a detritivore-based functioning relying essentially on spatial exports from more autotrophic systems²⁸.”*
- Lines 172-174: *“Moreover, even some terrestrial ecosystems can be net heterotrophic (around 20%), for instance due to perturbations like fires, harvesting, or drought, which could disrupt production capacity.”*

Third, we integrated the NEP figure in the main text, since it is now discussed in more detail, and we referred to it as part of the analysis, lines 69-70: *“We additionally compiled measurements of net ecosystem production to relate the importance of spatial flows to the degree of ecosystem heterotrophy.”*

6) Thank you for adding some details on the number of studies requiring unit conversions and addressing some of the assumptions of your conversion factors in this revised ms.

Authors’ response: Thank you again for your prior suggestions, they were very helpful.

Specific comments:

7) line 107-113. The wording here is awkward – consider rephrasing.

Authors’ response: We rephrased the sentence as follows, lines 112-115: *“These patterns result from freshwater and benthic ecosystems displaying low gross primary production (GPP) while receiving abundant cross-ecosystem inflows, which contrasts with terrestrial ecosystems receiving little inflows but producing abundant biomass.”*

8) line 117 and 120. Remove “a” in both cases.

Authors’ response: done accordingly.

9) lines 129-132. This is not new but the novelty lies in the huge scope of the data. I recommend the authors re-phrase this along the lines of “Our extensive synthesis of X confirms the long-held belief that ‘things flow down hills’ (i.e. Lindeman, Leroux)...” Phrasing it this way does not take away from the value of the work.

Authors’ response: We agree. We have now rephrased as suggested, and moved the sentence just after the first discussion section (in which we wanted to describe Figure 5), along with the proposed relevant references, lines 138-139: “*We thus provide a quantitative confirmation of the long-held belief that ‘things flow down hills’^{25,26}”.*

10) lines 132-134. I appreciate this new sentence but I think it would be better placed near the end in a section highlighting future directions. I think the subsidy point is made very well with lines 134-139.

Authors’ response: Thank you for your positive comment on this ‘subsidy point’. We expanded it with a sentence on net ecosystem production (see response to your point 5). Regarding the new sentence, we followed your suggestion to move it later in the text in a part on future directions, lines 211-215: “*An examination of the ultimate fate of spatial flows in recipient ecosystems was beyond the scope of our dataset, but next challenges for spatial ecology must include such assessments, notably considering quantity versus quality effects of spatial flows, as well as relating animal activities at different scales to fluxes of resources linking ecosystems⁴⁶.”*

11) line 138. Replace “does” with “must”.

Authors’ response: done accordingly.

12) lines 141-144. Nice example but consider adding specific details (study system, global synthesis, etc.). This paper has an impressive amount of data yet it is still very conceptual. Real examples with specific details will help ground it for the readers.

Authors’ response: We added details on this study as suggested, lines 159-162: “*Furthermore, the degree of integration into recipient food webs can depend on the characteristics of the carbon introduced there, with, for instance, terrestrial dissolved versus particulate organic carbon, respectively, subsidizing pelagic bacteria versus*

*benthic invertebrates and zooplankton in lakes*³⁰.” . Note that a lot of concrete examples from the data are also cited in the result section.

REVIEWERS' COMMENTS:

Reviewer #1 (Remarks to the Author):

Earlier I thought the revision of this paper was quite adequate, so I recommend acceptance of this version

Reviewer #2 (Remarks to the Author):

The authors have done a very comprehensive job of addressing all of the reviewer and editors concerns. I am particularly impressed by their multi-level statistical analysis of the NEP data and their incorporation of revised text pertaining to these results. The statistical analyses are simple but clever and provide robust support for their original findings. The new text pertaining to these results is well balanced and provides a nice discussion of the potential drivers of variation around NEP within and across ecosystems. I have no additional concerns with the ms and I congratulate the authors on a fine piece of work.

REVIEWERS' COMMENTS:

Reviewer #1 (Remarks to the Author):

Earlier I thought the revision of this paper was quite adequate, so I recommend acceptance of this version

Authors' response: Thank you very much for your comments and evaluation.

Reviewer #2 (Remarks to the Author):

The authors have done a very comprehensive job of addressing all of the reviewer and editors concerns. I am particularly impressed by their multi-level statistical analysis of the NEP data and their incorporation of revised text pertaining to these results. The statistical analyses are simple but clever and provide robust support for their original findings. The new text pertaining to these results is well balanced and provides a nice discussion of the potential drivers of variation around NEP within and across ecosystems. I have no additional concerns with the ms and I congratulate the authors on a fine piece of work.

Authors' response: Thanks a lot for your evaluation and encouragements.